

# Genome-wide identification and analysis of circular RNAs differentially expressed in the longissimus dorsi between Kazakh cattle and Xinjiang brown cattle

Xiang-Min Yan[1,2,*], Zhe Zhang[1,*], Yu Meng[1], Hong-Bo Li[2], Liang Gao[3], Dan Luo[1], Hao Jiang[1], Yan Gao[1], Bao Yuan[1] and Jia-Bao Zhang[1]

[1] Department of Laboratory Animals, Jilin University, Changchun, Jilin, China
[2] Institute of Animal Husbandry, Xinjiang Academy of Animal Husbandry, Ürümqi, Xinjiang, China
[3] Yili Vocational and Technical College, Yili, Xinjiang, China
* These authors contributed equally to this work.

Corresponding authors
Bao Yuan, yuan_bao@jlu.edu.cn
Jia-Bao Zhang, zjb@jlu.edu.cn

## ABSTRACT

Xinjiang brown cattle have better meat quality than Kazakh cattle. Circular RNAs (circRNAs) are a type of RNA that can participate in the regulation of gene transcription. Whether circRNAs are differentially expressed in the longissimus dorsi between these two types of cattle and whether differentially expressed circRNAs regulate muscle formation and differentiation are still unknown. In this study, we established two RNA-seq libraries, each of which consisted of three samples. A total of 5,177 circRNAs were identified in longissimus dorsi samples from Kazakh cattle and Xinjiang brown cattle using the Illumina platform, 46 of which were differentially expressed. Fifty-five Gene Ontology terms were significantly enriched, and 12 Kyoto Encyclopedia of Genes and Genomes pathways were identified for the differentially expressed genes. Muscle biological processes were associated with the origin genes of the differentially expressed circRNAs. In addition, we randomly selected six overexpressed circRNAs and compared their levels in longissimus dorsi tissue from Kazakh cattle and Xinjiang brown cattle using RT-qPCR. Furthermore, we predicted 66 interactions among 65 circRNAs and 14 miRNAs using miRanda and established a coexpression network. A few microRNAs known for their involvement in myoblast regulation, such as miR-133b and miR-664a, were identified in this network. Notably, bta_circ_03789_1 and bta_circ_05453_1 are potential miRNA sponges that may regulate insulin-like growth factor 1 receptor expression. These findings provide an important reference for prospective investigations of the role of circRNA in longissimus muscle growth and development. This study provides a theoretical basis for targeting circRNAs to improve beef quality and taste.

## INTRODUCTION

Circular RNAs (circRNAs) are a newly discovered class of RNAs that exist in the form of unique covalent rings with no 5′ caps or 3′ tails (*Memczak et al., 2013*). CircRNAs are

approximately 100 nucleotides (nt) in length. Because circRNAs usually have no poly-A tails, they exhibit greater stability and sequence conservation than normal linear RNA molecules (*Guo et al., 2014*). CircRNAs have many biological characteristics, such as extensive expression, tissue specificity, high conservation, and cell specificity. CircRNAs also have many regulatory functions, including interaction with RNA-binding proteins (*Conn et al., 2015*), regulation of parental gene transcription (*Li et al., 2015*), and sponging of microRNAs (miRNAs) (*Hansen et al., 2013*).

With improvements in living standards, Chinese residents have paid increasing attention to the quality of beef. Due to the limitations of Kazakh cattle, we introduced Swiss brown cattle into the lineage to form Xinjiang brown cattle. Fatty acid composition not only determines the toughness/fatness of adipose tissue and the oxidative stability of muscles but also affects the taste of meat and the color of muscle tissue (*Wood et al., 2008*). *Li et al. (2017)* studied circRNAs in the longissimus dorsi muscle of sheep before and after delivery using RNA-seq. *Heumüller & Dimmeler (2019)* revealed that circRNAs control the functions of vascular smooth muscle cells in mice. Furthermore, circRNAs have recently been shown to play vital roles in cell proliferation, differentiation, autophagy and apoptosis during development. However, no report has described the association between muscle development and circRNA expression in Xinjiang brown cattle.

According to many studies, circRNAs function as miRNA sponges (*Memczak et al., 2013*). For example, CircHIPK3 promotes colorectal cancer growth and metastasis by sponging miR-7 to regulate insulin-like growth factor 1 receptor (IGF1R) expression (*Zeng et al., 2018*). In addition, during osteogenesis, circUSP45 inhibits glucocorticosteroid-induced femoral head necrosis by sponging miR-127-5p through the PTEN/AKT serine/threonine kinase 1 (AKT) signaling pathway (*Kuang et al., 2019*). Moreover, Circ-8073 regulates CEP55 expression by sponging miR-449a and promotes the proliferation of goat endometrial epithelial cells through a mechanism mediated by the PI3K/AKT/mTOR pathway (*Liu et al., 2018*). However, no study has constructed a circRNA-miRNA-mRNA regulatory network for Xinjiang brown cattle.

In this study, we systematically investigated the circRNA levels in longissimus dorsi tissue from Kazakh cattle and Xinjiang brown cattle using RNA-seq. In addition, we predicted the interactions between miRNAs and circRNAs. Our findings will provide a meaningful resource for more profound investigations of the regulatory functions of circRNAs in cattle and will contribute to a better understanding of muscle growth and development in mammals.

## MATERIALS AND METHODS

### Ethics statement

This experiment was performed in strict accordance with the guiding principles of the guidelines for the care and use of experimental animals at Jilin University. All experimental programs were approved by the Animal Care and Use Committee of Jilin University (license number: 201809041).

## Animal and tissue preparation

Kazakh cattle and Xinjiang brown cattle were provided by the Xinjiang Yili Yixin Cattle and Sheep Breeding Cooperative. After cattle were slaughtered in accordance with the procedure of the slaughterhouse, and the longissimus muscle was collected at the slaughter line. We tested six longissimus dorsi: three from Kazakh cattle and three from Xinjiang brown cattle. We chose 30-month-old adult bullocks weighing approximately 600 kg. All of the samples were immediately snap-frozen in liquid nitrogen and stored at −80 °C until RNA extraction.

## Hematoxylin-eosin staining

Histological observations were performed using conventional histological methods after preparing longissimus dorsi muscle tissues from Kazakh cattle and Xinjiang brown cattle that had been preserved with 4% paraformaldehyde for 72 h. Hematoxylin-eosin staining was performed (*Guardiola et al., 2017*). The morphology of the muscle tissue was observed using a fluorescence microscope (Olympus, Japan).

## Total RNA isolation

Total RNA was extracted from each group (the Kazakh cattle group and the Xinjiang brown cattle group) using TRIzol (Invitrogen, NY, USA). A NanoDrop 2000 spectrophotometer (Thermo Fisher Scientific, Waltham, MA, USA) was used to evaluate the concentrations and quality of the RNA, and agarose gel electrophoresis was used to evaluate the integrity of the RNA (*Fu et al., 2018*).

## RNA library construction

Equal amounts of RNA (1 μg of RNA) from each sample were used to construct the circRNA libraries. The mRNA was enriched with magnetic mRNA Capture Beads, purified using DNA Clean Beads and fragmented (with the addition of First-Strand Synthesis Reaction Buffer and random primers). Different index tags were selected for library construction in accordance with the instructions of the NEBNext® Ultra™ RNA Library Prep Kit for the Illumina platform (NEB, Ipswich, MA, USA) (*Pang et al., 2019*). The RNA was cut into short fragments by adding fragmentation buffer to the reaction system. Six-base random primers (random hexamers) were added to synthesize the first strand of the cDNA, and buffers, dNTPs, RNase H and DNA polymerase I were added to synthesize the second strand of the cDNA. The double-stranded cDNA products were purified. End Repair Reaction Buffer and End Repair Enzyme Mix were added to the purified products, and the tubes were placed in a PCR instrument to perform the reactions (*Xia et al., 2016*). We conducted paired-end sequencing with a read length of 150 bp. The different libraries were sequenced with an Illumina NovaSeq 6000 platform by BioMarker Technologies (Beijing, China) based on the target machine data.

## Sequencing quality control

The obtained raw data containing linker sequences and low-quality sequences were subjected to quality control protocols to ensure accurate analysis. Processing of the data

produced high-quality sequences (clean reads). We removed the reads containing linker sequences and the low-quality reads to ensure data quality. We also deleted sequences with >5% N bases (uncertain bases). The clean data were aligned with the specified reference genome to obtain mapped data. The Q30 value was used as the standard for testing the quality of our library (*Zhang et al., 2019a*).

## Identification of circRNAs

CircRNAs were predicted with the CIRI and find_circ software packages (*Zhang et al., 2019b*). The circBase database includes circRNA sequences from five organisms: humans, mice, coelacanths, fruit flies and nematodes. Since the experimental samples were derived from cattle, we predicted the circRNAs using CIRI software (*Gao, Wang & Zhao, 2015*). In addition, find_circ was used since the circRNA loci were not able to be directly aligned with the genome; find_circ anchors independent reads with the 20 base pairs at the end that are incompatible with the genome to match the reference genome with only matching sites (*Memczak et al., 2013*). We downloaded the *Bos taurus* reference genome from the Ensembl genome browser (http://www.ensembl.org/Bos_taurus/Info/Index) (*Zhou et al., 2015*). If the two anchors were aligned in reverse directions in the linear region, the anchor reads were extended until a circRNA link was detected. This sequence was considered the circRNA sequence.

## Differential expression analysis

The circRNA expression in each sample was determined and is presented as the number of transcripts per million kilobases (*Zhou et al., 2010*). The differential expression of circRNAs was analyzed with DEseq (*Bao et al., 2019*; *Love, Huber & Anders, 2014*). In the differential expression analysis, a fold change (FC) ≥ 1.5 served as the screening criterion. The FC indicates the ratio of the expression levels between two samples (groups). As an independent statistical hypothesis test for circRNA expression levels, the differential expression analysis of circRNAs tended to produce false positive results. Therefore, in this study, the Benjamini-Hochberg correction was used. The original $P$-values were analyzed, and false discovery rates (FDRs) were used as the pivotal indicators to screen differentially expressed circRNAs.

## Target site prediction and functional enrichment analysis

A circRNA-miRNA-mRNA coexpression network was established according to the miRNA binding sites predicted by miRanda (http://www.microrna.org/microrna/home.do) (*Betel et al., 2010*; *Liu et al., 2019*). TargetScan was used to predict the binding sites for miRNAs in mRNAs (*Agarwal et al., 2015*). According to the mapped data, the high-quality sequencing results were subjected to Kyoto Encyclopedia of Genes and Genomes (KEGG) circRNA analysis, circRNA binding site analysis, circRNA gene analysis, differential circRNA expression analysis, and Gene Ontology (GO) circRNA gene analysis. After the circRNA mapping and miRanda analyses, the names of the circRNA target genes were subjected to GO analysis using the top GO R packages (*Fedorova et al., 2019*). KEGG enrichment was performed using KOBAS software to analyze the circRNA target genes

(*Mao et al., 2005*; *Xie et al., 2011*). GO terms and KEGG pathways for which $P < 0.05$ were considered significantly enriched.

## Quantitative real-time PCR (RT-qPCR) analysis of circRNAs

To further detect the differentially expressed circRNAs between the treatment groups, SuperReal PreMix Plus (SYBR Green) (Tiangen, China) was used to perform RT-qPCR according to the manufacturer's instructions (*Fu et al., 2018*; *Han et al., 2019*). The levels of the circRNAs were determined relative to the expression levels of β-*actin*. RT-qPCR was performed using the following reaction system: 10 μL of 1× SYBR Premix DimerEraser, 1 μL of cDNA, 0.5 μL of upstream and downstream primers, and 8 μL of ddH$_2$O without RNase. The results were normalized to β-*actin* expression. The relative expression levels of the circRNAs were determined with the $2^{-\Delta\Delta CT}$ method based on the cycle threshold (Ct) values (Table S1).

## Statistical analysis

The data are presented as the mean ± SD from three independent experiments in RT-qPCR analysis. The data were analyzed with SPSS 23.0 software. One-way ANOVA was used to determine the significance of differences, and $P < 0.05$ was considered to indicate a significant difference.

# RESULTS

## Morphology of the longissimus dorsi in Kazakh cattle and Xinjiang brown cattle

Compared with Kazakh cattle, Xinjiang brown cattle exhibited dramatic differences in longissimus dorsi morphology (Figs. 1A and 1B). We compared the number, area, diameter and density of muscle fibers in the longissimus dorsi between these breeds and found a greater number of muscle fibers in Kazakh cattle tissue than in Xinjiang brown cattle tissue. However, the area and diameter of muscle fibers in Kazakh cattle were smaller than those in Xinjiang brown cattle. No difference in the density of muscle fibers was observed between Kazakh cattle and Xinjiang brown cattle (Fig. 1C).

## Overview of circRNA sequencing data

We established two RNA-seq libraries. The libraries were sequenced with an Illumina NovaSeq6000 platform and then subjected to a rigorous filtering pipeline. Before circRNA identification, quality control was carried out by calculating the Q30 value and the GC content (Table S2). Ultimately, we obtained 5,177 circRNAs from the RNA-seq data. We detected 22,677 genes and 929 differentially expressed genes. Among the differentially expressed genes, 471 genes were upregulated, and 458 genes were downregulated. Moreover, the 5,177 circRNAs were distributed on 29 autosomes and the X chromosome. Chromosome 2 contained the most circRNAs, and chromosome 27 contained the fewest circRNAs (Fig. 2A). Next, we analyzed the genomic origins of the expressed circRNAs. The sizes of the circRNA candidates ranged from 200 nt to >2000 nt, but the lengths of most of the candidates were between 400 nt and 800 nt. Approximately 74.85% of the

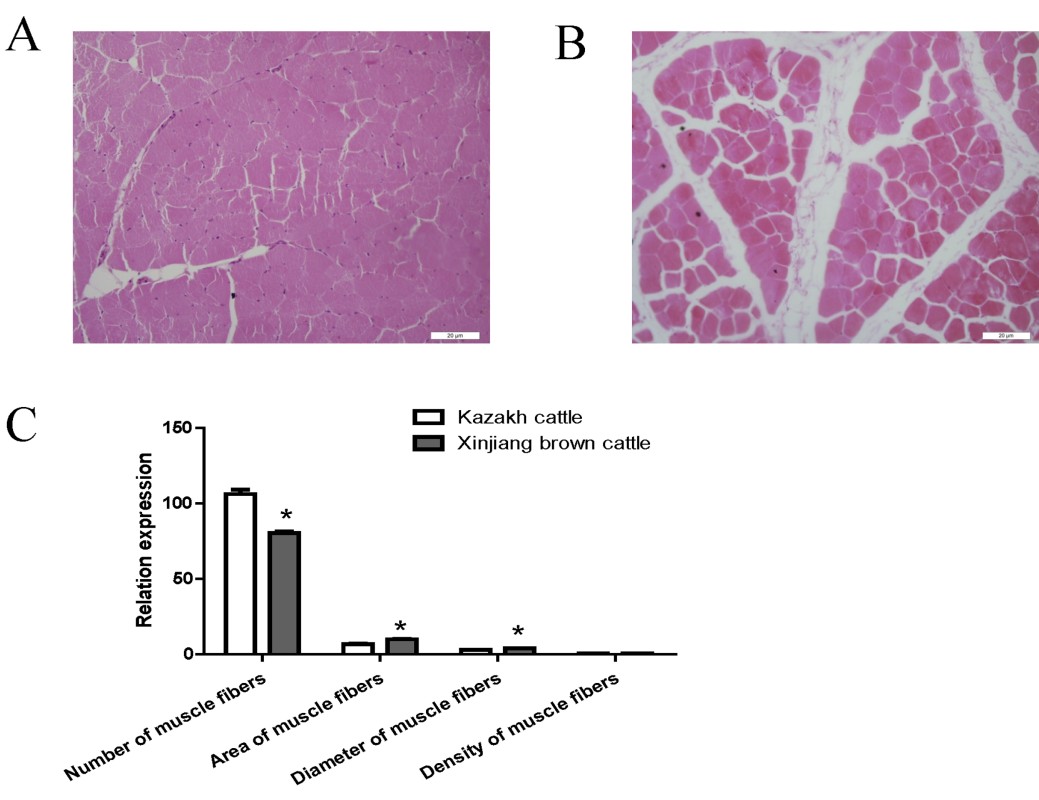

**Figure 1 The morphology of the longissimus dorsi differed between Kazakh cattle and Xinjiang brown cattle.** (A and B) Tissue morphology of the longissimus muscle in Kazakh cattle and Xinjiang brown cattle, respectively. (C) Comparisons of the number, area, diameter and density of muscle fibers in the longissimus dorsi between Kazakh cattle and Xinjiang brown cattle. "*" indicates significant differences.

circRNAs had a predicted spliced length of less than 2000 nt, whereas circRNAs with lengths greater than 2000 nt accounted for 25.15% of the circRNAs (Fig. 2B).

## Identification of differentially expressed circRNAs

A volcano plot was constructed to display the relation between the FDR and the FC values for the levels of all circRNAs and thus to quickly reveal the differences in circRNA expression patterns (and their statistical significance) between the two libraries (Fig. 3A). An MA map was constructed to display the overall distribution of the expression levels and the FCs in circRNA expression between the two libraries (Fig. 3B). We identified 46 circRNAs that were differentially expressed in the longissimus dorsi muscle between Kazakh cattle and Xinjiang brown cattle (Table S3). The differentially expressed circRNAs included 26 upregulated and 20 downregulated circRNAs in Xinjiang brown cattle compared to Kazakh cattle. We examined the expression patterns of the differentially expressed circRNAs using a systematic cluster analysis to explore the similarities and differences between Kazakh cattle and Xinjiang brown cattle (Fig. 3C).

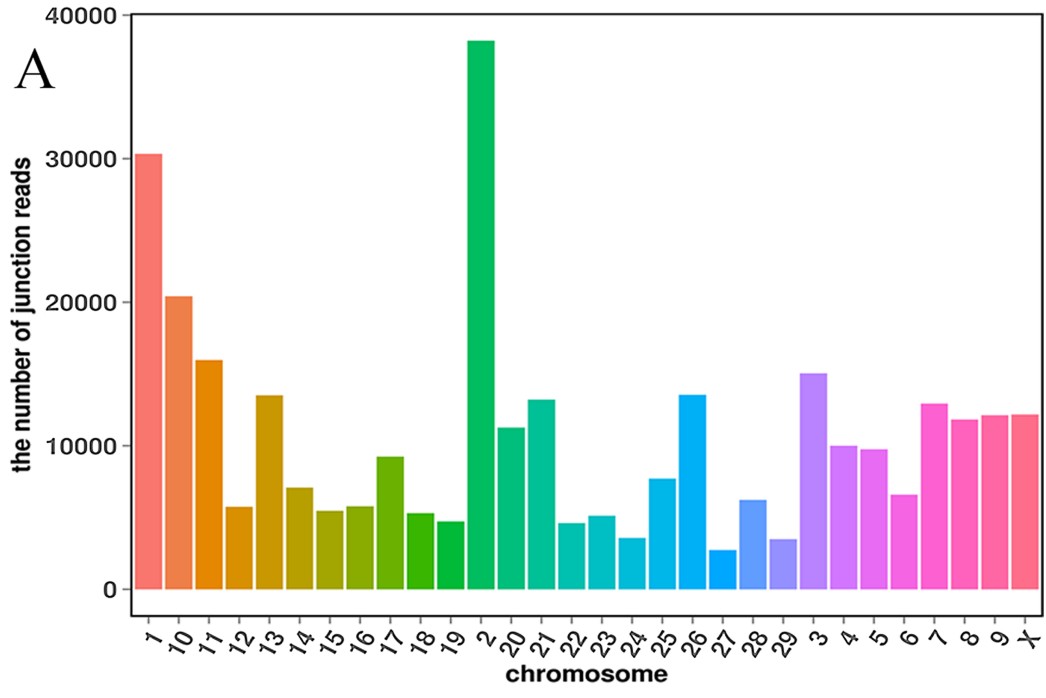

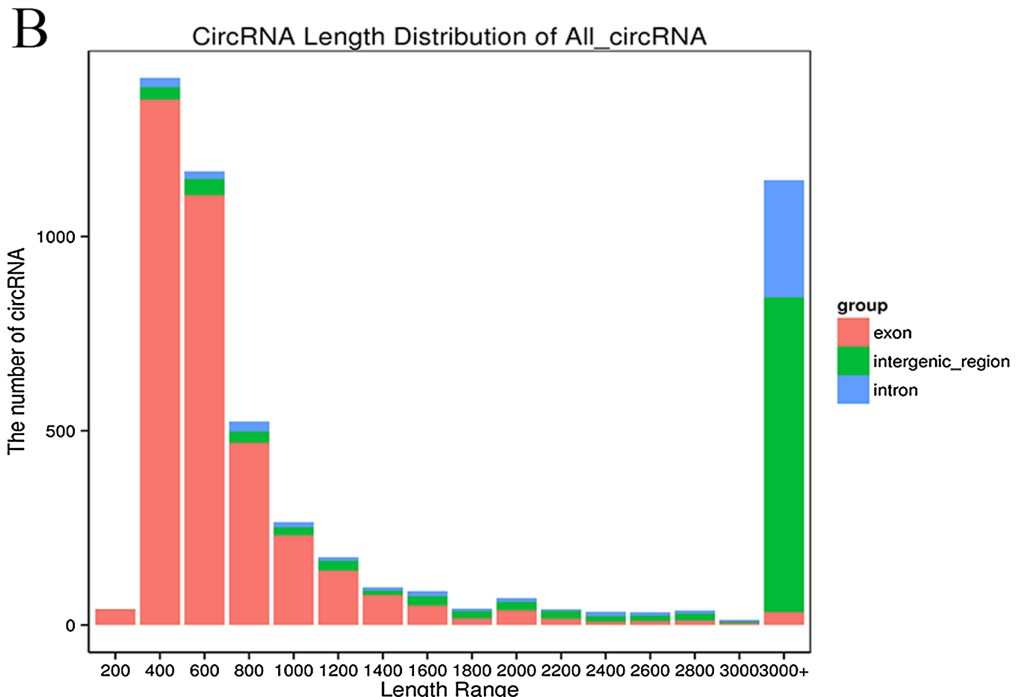

**Figure 2** **Deep sequencing of circRNAs in Kazakh cattle and Xinjiang brown cattle.** (A) Distribution of the circRNAs on the cattle chromosomes. (B) Distributions of the sequence lengths of the circRNAs.

## Enrichment of the differentially expressed circRNAs

Gene Ontology and KEGG pathway enrichment analyses were conducted to analyze the enriched terms and pathways associated with the differentially expressed circRNAs.

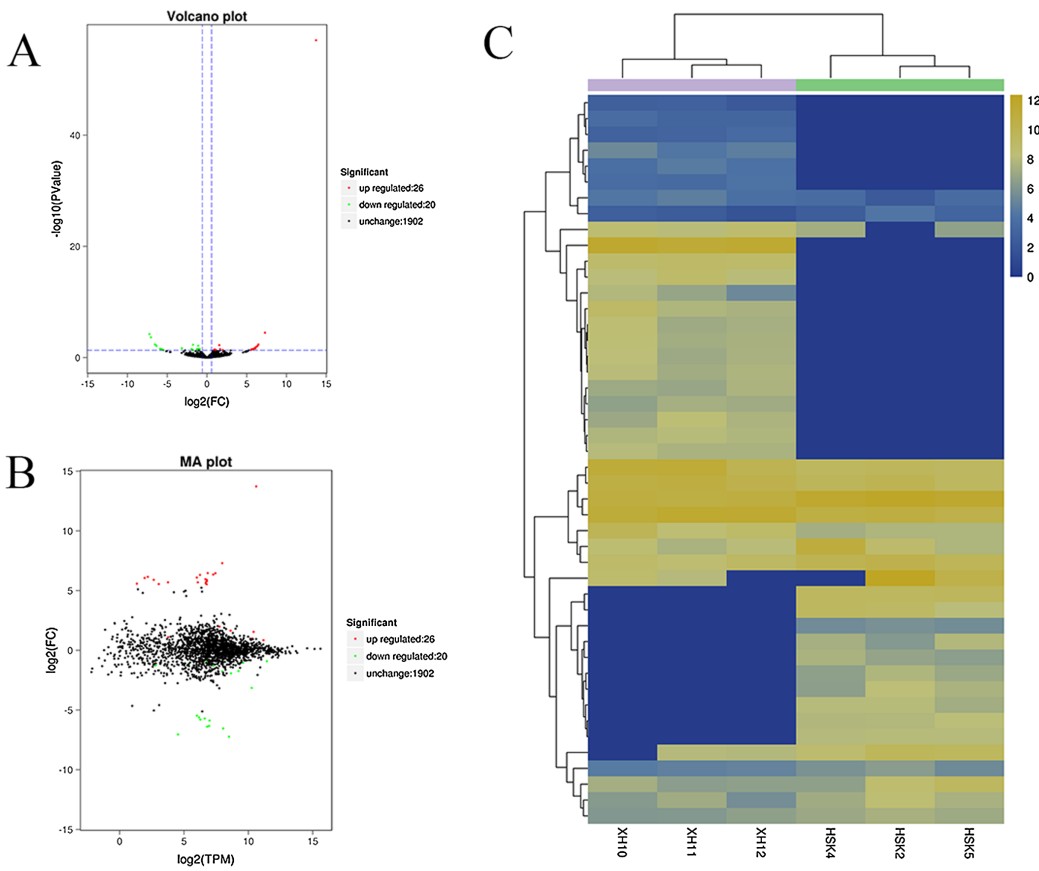

**Figure 3** **Differentially expressed circRNAs were identified.** (A) The volcano plot and (B) the MA map show the circRNAs that were differentially expressed in the longissimus muscle between Kazakh cattle and Xinjiang brown cattle. The red dots indicate upregulated genes, while the green dots indicate downregulated genes. (C) Analysis of the expression patterns of the differentially expressed circRNAs. The highest to lowest fold changes are indicated with a color code ranging from red to green, respectively.                                 

GO annotation was performed to obtain information about the functions of the differentially expressed circRNAs. The genes generating the circRNAs were annotated in three GO categories: molecular function, cellular component, and biological process. According to the GO analysis, 55 GO terms were significantly enriched and were mainly associated with the cell part (GO:0044464), binding (GO:005488) and cellular process (GO:0009987) terms (Table S4). Figure 4 shows the GO annotations for the upregulated and downregulated mRNAs in the cellular component, biological process and molecular function categories. In addition, 12 KEGG pathways contained differentially expressed genes, including mTOR signaling pathways, TGF-beta signaling pathways, and Hippo signaling pathways (Table S5). Thus, the differentially expressed circRNAs might function as important regulators of muscle growth and development.

## CircRNA-miRNA-mRNA network

CircRNAs can act as competing endogenous RNAs (ceRNAs) by functioning as miRNA sponges; therefore, we searched the sequences of the differentially expressed circRNAs and

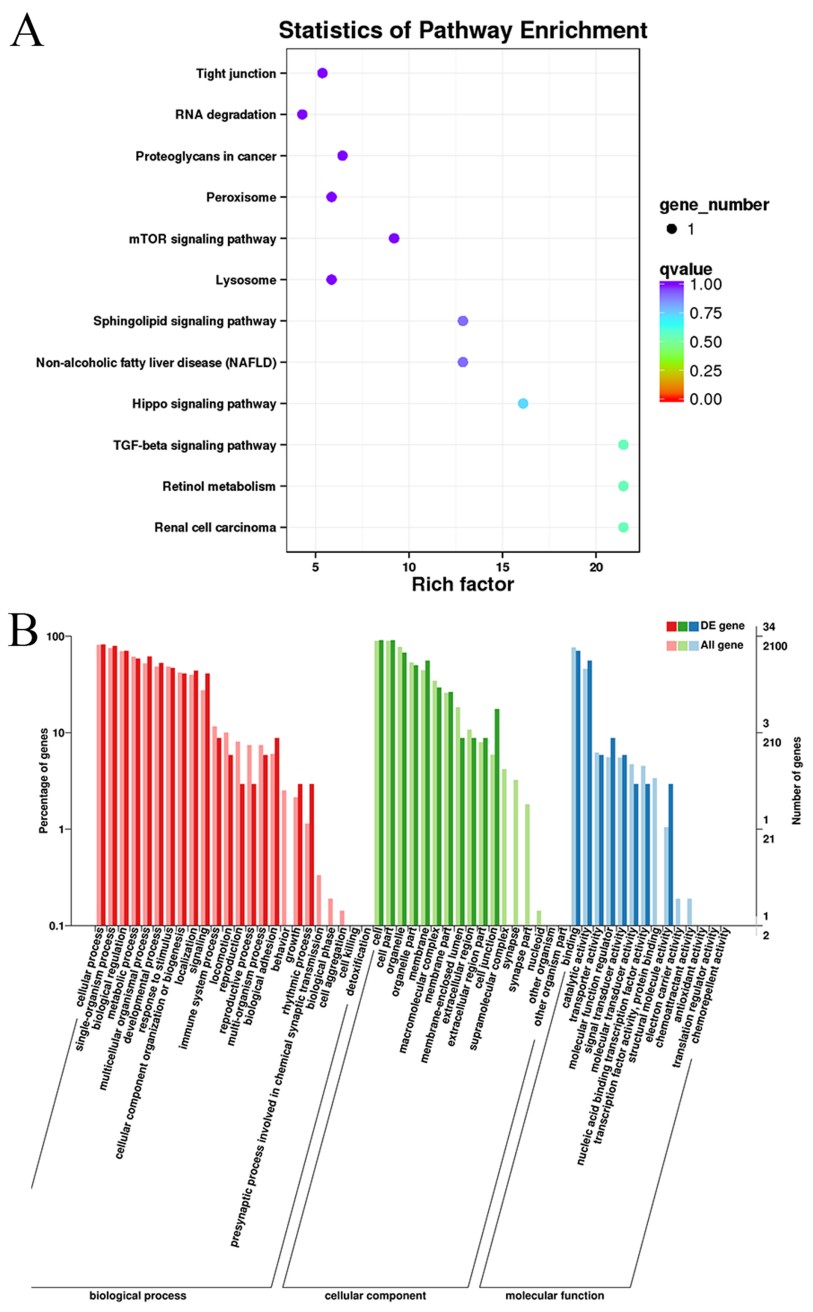

**Figure 4 KEGG and GO enrichment analyses of target genes in muscle tissue.** (A) Scatter plot of the enriched KEGG pathways for the differentially expressed circRNA cistarget genes. (B) Diagram of the GO annotations for the differentially expressed circRNA target genes. The abscissa indicates the GO classification, the left ordinate indicates the percentage of all circRNA target genes, and the right ordinate indicates the number of circRNA target genes.

established an interactive network map. We predicted 14 miRNAs that may target the 3′ untranslated region of IGF1R. We predicted the interactions between circRNAs and miRNAs with miRanda to further analyze the functions of the circRNAs. Then, we established the interactive network map; the network included 66 relationships in which

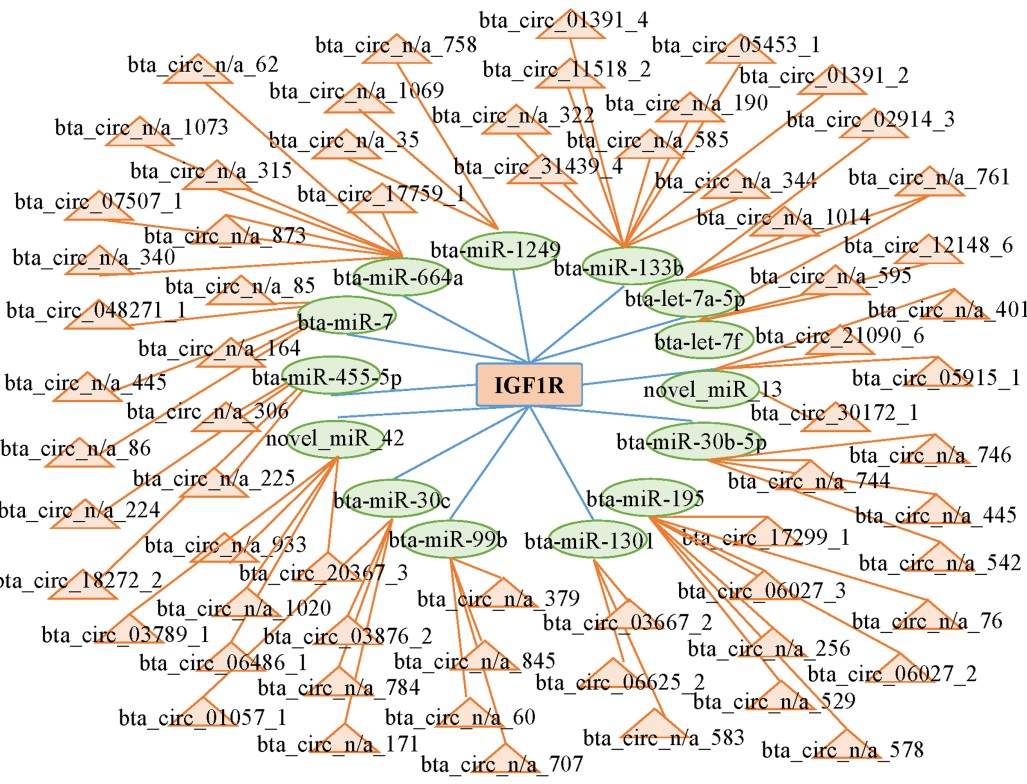

**Figure 5 Network of interactions between circRNAs and miRNAs based on the miRanda program.**

65 circRNAs interacted with 14 miRNAs (Fig. 5; Table S6). We focused on some extensively studied miRNAs in the network that play crucial roles in muscle growth and development, such as miR-664a and miR-133b. Using the miRNAs that were closely associated with muscle growth and development, we identified relevant candidate circRNAs that may also be involved in these processes.

## Validation of highly expressed circRNAs and two key circRNAs

We randomly selected six circRNA candidates and designed primers spanning the junction areas to confirm the reproducibility of the circRNA data acquired from RNA-seq analysis. The six differentially expressed circRNAs between the two RNA-seq libraries included three upregulated circRNAs (bta_circ_06771_2, bta_circ_19409_2 and bta_circ_12705_1) and three downregulated circRNAs (bta_circ_01274_2, bta_circ_11905_4 and bta_circ_06819_5). The results were highly consistent with the RNA-seq results (Fig. 6A). Moreover, we detected the expression of bta_circ_03789_1 and bta_circ_05453_1 in longissimus dorsi from Kazakh cattle and Xinjiang brown cattle, and the results indicated that both circRNAs were upregulated in Xinjiang brown cattle compared to Kazakh cattle. Based on these results, the trends in the expression of the two circRNAs were consistent with the expression of IGF1R mRNA (Figs. 6B–6D). Therefore, bta_circ_03789_1 and bta_circ_05453_1 may be miRNA sponges that regulate the IGF1R gene and further affect the regulation of related factors in the longissimus dorsi muscle in cattle.

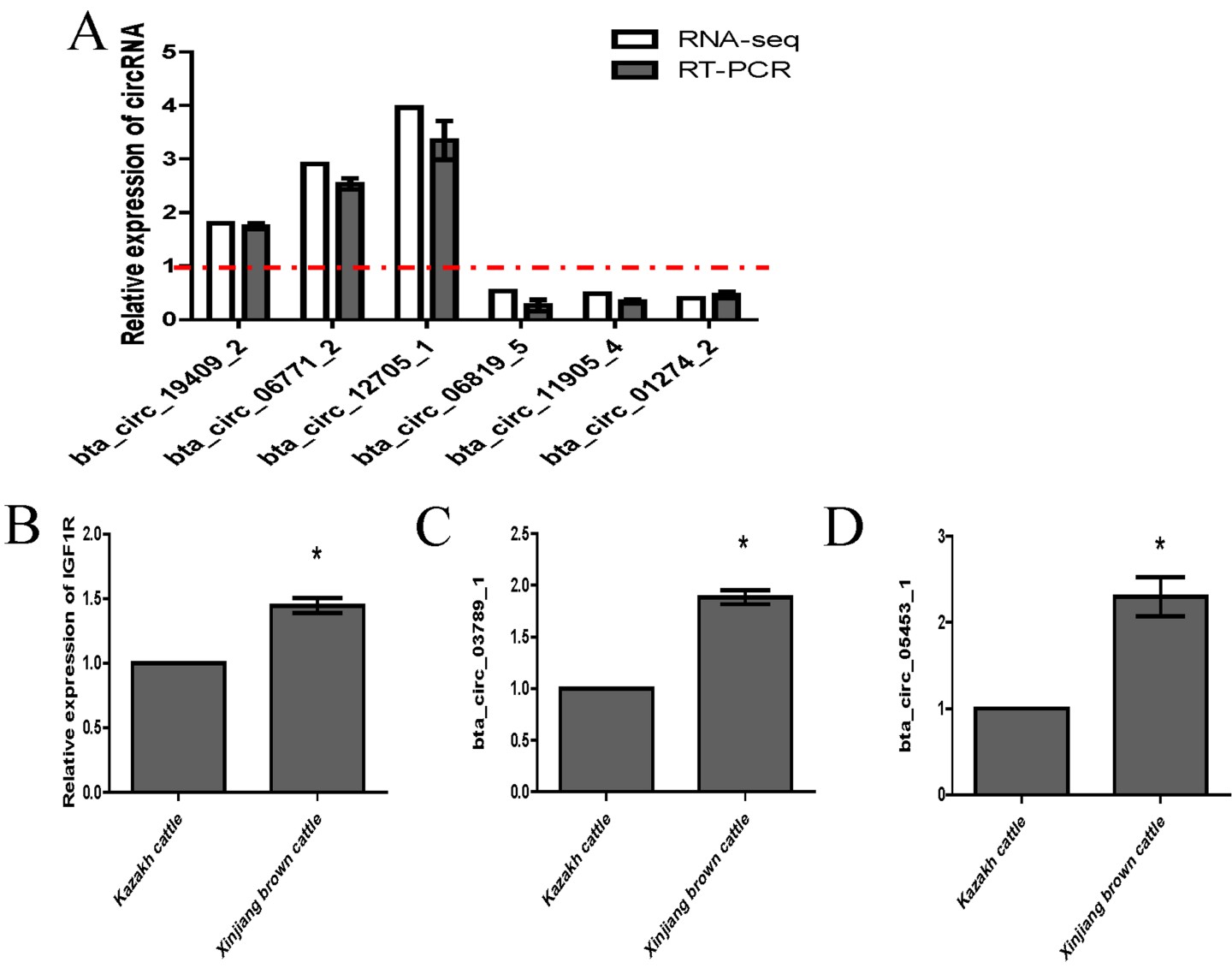

**Figure 6 Validation of highly expressed circRNAs and two pivotal circRNAs.** (A) Expression of negatively regulated and positively regulated circRNAs in the longissimus dorsi muscle in Kazakh cattle and Xinjiang brown cattle. (B–D) Expression levels of IGF1R, circ_03789_1 and circ_05453_1 in the longissimus dorsi muscle in Kazakh cattle and Xinjiang brown cattle. All experiments were repeated more than three times. The data are presented as the means ± SDs. Statistical significance was analyzed using one-way ANOVA, and "*" indicates significant differences.

## DISCUSSION

Beef quality has become increasingly important with improvements in living standards. Notably, *Nolte et al. (2019)* identified a biological network of lncRNAs associated with metabolic efficiency in cattle. In addition, *Ma et al. (2019)* have proposed that IGF1R copy number variation (CNV) is a molecular marker that can be used to improve the production of beef during cattle breeding. *Apaoblaza et al. (2019)* compared the muscle energy of grass-fed and grain-fed cattle and found that grass-fed beef had higher levels of enzymes reflective of oxidative metabolism. Furthermore, Low expression of MyHC-IIa

has been observed in tough meat relative to tender meat, and MyHC-IIa is considered to be a biomarker of meat quality (*Chardulo et al., 2019*). *Zhang et al. (2018)* evaluated the kinetics of circRNA expression in C2C12 myoblasts using RNA-seq data. Similarly, *Cao et al. (2018b)* investigated the expression profiles of circRNAs in sheep striated skeletal muscle. However, no report has described the association between muscle development and circRNA expression in Xinjiang brown cattle.

CircRNAs, which area newly discovered type of RNA, form covalently closed continuous rings and are expressed at high levels in eukaryotic transcriptomes (*Qu et al., 2015*). CircRNAs have been reported to be relevant to cardiovascular diseases (*Fan et al., 2017*), cell senescence (*Cai et al., 2019*), diabetes (*Tian et al., 2018*), regenerative medicine (*Cao et al., 2018a*) and cancer (*He et al., 2017*). However, no studies have examined the expression of circRNAs associated with muscle development in Xinjiang brown cattle. In this study, we identified 5177 circRNAs in longissimus dorsi tissuesfrom Kazakh cattle and Xinjiang brown cattle using RNA-seq data. We identified 46 circRNAs that were differentially expressed in the longissimus dorsi muscle between these two breeds. Furthermore, we identified 55 significant GO terms and 12 meaningful KEGG pathways. The KEGG pathways were associated with mTOR signaling pathways, TGF-beta signaling pathways, and Hippo signaling pathways. Compared to Kazakh cattle, Xinjiang brown cattle have strong adaptability and disease resistance and excellent meat quality (*Agarwal et al., 2015*). Whether the identified differentially expressed circRNAs affect muscle generation and differentiation via the identified signaling pathways will be the focus of our next study. Many studies have reported that circRNAs act via related pathways to affect the development and production of muscle. As shown in a study by *Jin et al. (2017)*, Lnc133b functions as a molecular sponge of miR-133b to regulate the expression of IGF1R, promoting satellite cell proliferation and repressing cell differentiation (*Jin et al., 2017*).

CircRNAs and mRNAs have similar sequences that are bound by the same miRNAs. When bound by miRNAs, upregulated circRNAs serve as ceRNAs that prevent the miRNAs from binding to their mRNA targets and thus promote the expression of mRNAs at the posttranscriptional level. In this study, we predicted 66 interactions among circRNAs and miRNAs in longissimus dorsi muscle from Kazakh cattle and Xinjiang brown cattle. Among the interacting molecules, bta_circ_03789_1 and bta_circ_05453_1 were differentially expressed circRNAs that were determined to act as sponges; bta_circ_03789_1 was predicted to sponge miR-664a, while bta_circ_05453_1 was predicted to sponge both miR-7. Some research has shown that miR-664 promotes myoblast proliferation and inhibits myoblast differentiation by targeting SRF and Wnt1 (*Cai et al., 2018*). Differential expression of miR-7 has been observed in myoblasts from subjects with facioscapulohumeral muscular dystrophy and in control primary myoblasts (*Dmitriev et al., 2013*). In this study, we created a catalog of circRNAs expressed in the longissimus dorsi and identified differentially expressed circRNAs between Kazakh cattle and Xinjiang brown cattle. Furthermore, we predicted two circRNAs that function as miRNA sponges and potentially regulate the expression of the IGF1R gene to subsequently regulate muscle growth and development. Our study provides an important resource

for understanding circRNA biology in the contexts of genetics and breeding and provides insights into the functions of circRNAs in muscle.

## CONCLUSION

These data jointly reveal significant differences in the expression of circRNAs in the longissimus dorsi between Kazakh cattle and Xinjiang brown cattle. In the future, we will study how the differentially expressed circRNAs regulate muscle growth and development. Our findings will provide a meaningful resource for more profound investigations of the regulatory functions of circRNAs in cattle longissimus muscle growth and development. This study provides a theoretical basis for targeting circRNAs to improve beef quality and taste.

### Funding

This work was supported by the National Key Research and Development Program of China (2018YFD0501801), the Modern Agricultural Industrial Technology System (CARS-38) and the Xinjiang Autonomous Region basic research business fee (KY2019117). The funders had no role in study design, data collection and analysis, decision to publish, or preparation of the manuscript.

### Grant Disclosures

The following grant information was disclosed by the authors:
National Key Research and Development Program of China: 2018YFD0501801.
Modern Agricultural Industrial Technology System: CARS-38.
Xinjiang Autonomous Region basic research business fee: KY2019117.

### Competing Interests

The authors declare that they have no competing interests.

### Author Contributions

- Xiang-Min Yan performed the experiments, analyzed the data, prepared figures and/or tables, authored or reviewed drafts of the paper, and approved the final draft.
- Zhe Zhang performed the experiments, analyzed the data, prepared figures and/or tables, authored or reviewed drafts of the paper, and approved the final draft.
- Yu Meng performed the experiments, authored or reviewed drafts of the paper, and approved the final draft.
- Hong-Bo Li performed the experiments, authored or reviewed drafts of the paper, and approved the final draft.
- Liang Gao performed the experiments, authored or reviewed drafts of the paper, and approved the final draft.
- Dan Luo analyzed the data, prepared figures and/or tables, and approved the final draft.
- Hao Jiang analyzed the data, prepared figures and/or tables, and approved the final draft.
- Yan Gao analyzed the data, prepared figures and/or tables, and approved the final draft.

- Bao Yuan conceived and designed the experiments, authored or reviewed drafts of the paper, and approved the final draft.
- Jia-Bao Zhang conceived and designed the experiments, authored or reviewed drafts of the paper, and approved the final draft.

## Animal Ethics

The following information was supplied relating to ethical approvals (i.e., approving body and any reference numbers):

The Animal Care and Use Committee of Jilin University approved the study (license number: 201809041).

## Data Availability

The RNA-seq data are available at NCBI BioProject, accession number: PRJNA532321.

## Supplemental Information

Supplemental information for this article can be found online at http://dx.doi.org/10.7717/peerj.8646#supplemental-information.

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
