# Peer review of "Genome-wide identification and analysis of circular RNAs differentially expressed in the longissimus dorsi between Kazakh cattle and Xinjiang brown cattle"

_PeerJ, doi:10.7717/peerj.8646_

## Round 0.1 · original submission · Major Revisions

Dear authors,

I received now comments from three independent reviewers. In view of the criticisms of both reviewers, a major revision is required in your manuscript. I ask you to revise your manuscript for all the comments the reviewers highlighted point by point.

Kind regards
The editor

Reviewer 1 ·

Basic reporting

The English language should be improved. Some examples where the language could be improved include lines 83-85 and the Conclusion section. Moreover, authors need to add a space between the word and the reference. Please, check line 67; 69; 76; 90; 100 and throughout the manuscript.

Experimental design

The study was performed using three samples of each cattle breed. While the sample numbers are on the lower side, this data could provide to research community interesting results about circRNA and the regulation of gene expression in skeletal muscle tissue.

Line 136. Authors need to add a reference for histological methods.

Line 140. These cattle groups were compared or not? Please, explain.

Line 207. Please, explain. Authors should specify which experiments were considered. Moreover, statistical analysis is poorly described.

Validity of the findings

The objectives and Conclusion need to be improved. The objective section should be more specific. Conclusion needs to be rewritten.

Line 314-315. What is the application of this results? Are those cricRNAs associated with meat quality traits or tissue development? Please, explain.

Line 338. This phenotype (longissimus muscle area) should be associated with circRNA data and the possible differences in gene expression. Thus, authors need to discuss how the circRNA regulate this muscle traits.

Additional comments

Lines 83-85. This sentence should be improved. Is this the objective of the current study (investigation of molecular mechanisms that control meat quality traits in Chinese brown cattle?) If this is the case, the Introduction section should be rewritten.

Annotated reviews are not available for download in order to protect the identity of reviewers who chose to remain anonymous.

Reviewer 2 ·

Basic reporting

The manuscript is well reported, it is a topical work and there are very interesting results.
The paper is technically well presented and written. I suggest to carefully revise some paragraphs for English.

Experimental design

Materiel and methods are well explained but some modifications should be completed
I am wondering how the histological was performed. I suggest to the authors to carefully check the whole M&M part and provide all the necessary and detailed information of the protocols used. I also recommend to the authors, one the protocols are described elsewhere to just detail them as supplementary data.
Line 139: RNA library construction and sequencing
Please add sub-title to explain the different steps of the method and add reference to this method
Line 197: RT-qPCR analysis of circRNAs
Please add reference to this method.
The statistical method is poorly described. I ask the authors to carefully check this part and describe step by step all the statistical analyses conducted in this manuscript. Also, the packages uses are to be given in the supplementary data as recommended by the journal.
Results are welled presented, but there are some repetitions between methods and results (line 222-225) and (line 240-246)
To clarify the main message of the discussion section of this manuscript, I suggest to the authors to add sub-sections with informative titles and discuss accordingly.

Validity of the findings

The conclusion needs to be completed to give the take-home messages of this work. It is very poor and the authors are asked to highlight the novelty of their work and what are the future investigations.

Additional comments

This paper introduces interesting results. It needs substantial revision before I recommend it for PeerJ journal.

Reviewer 3 ·

Basic reporting

First of all, I would like to remark the importance of the paper, to present novel information regarding circRNAs. I have some general comments to make before moving on to a number of more specific items:
:
1- Tittle of the paper: it may need to be changed in order to fit better with the research results, which reveals significant difference in the expression of circRNA between two different cattle breeds.
2- I suggest to condense the introduction, most of the text is repeated in the discussion. It will be very interesting to hypothesize about environmental rear conditions of these breeds, and how it modulate the expression of circRNA. Also, materials and methods must be specified.
The “state of the art” should be emphasized in cattle and muscle mechanism of regulations.
3- Discussion should focus on results. It should propose an explanation about the differential expression of circRNA in the muscle of two different breeds. specially in myoblast mechanisms of regulations.

Specific points:
39-40 Muscle was the only tissue used, specific longisimus dorsi, that means only one sample, of 6 different animals (replicates) that need to be clarified, in the text and also in material and methods.
46- 48 Be specific about if the expression on paternal gene mean imprinted genes.
100-101 It is important to specify in which species circRNAs control the functions of vascular smooth muscle.
105- That asseveration need a reference.

Experimental design

Experimental are well design, was sequentially organized and responde to the main research aim. It has to be noted however, that the the researcher do no mention a control. Its is importante to have a values of reference about the circRNA expression in other tissue, or other species.

Validity of the findings

As the main result of the paper was to propose a possible role for two circRNA, which could be acting as a sponge of unknown miRNAs, and regulated the IGF1R gene expression, authors should propose next experimental design that shed light on the real functions of these circRNAs.

Additional comments

Please check the lines: 72, 251, 252, 313, 318, 329 to separate words. Check the english redation in 42-44

---

## Round 0.2 · accepted · Accept

Dear authors,

I am glad to inform you that your paper is now accepted for publication in PeerJ.

Kind regards,
Mohammed Gagaoua